# Global Attention Super-Resolution Algorithm for Nature Image Edge Enhancement

Zhihao Zhang , Zhitong Su, Wei Song and Keqing Ning *

Information Institute, North China University of Technology, Beijing 100144, China
* Correspondence: ningkq@ncut.edu.cn

**Abstract:** Single-image super-resolution (SR) has long been a research hotspot in computer vision, playing a crucial role in practical applications such as medical imaging, public security and remote sensing imagery. However, all currently available methods focus on reconstructing texture details, resulting in blurred edges and incomplete structures in the reconstructed images. To address this problem, an edge-enhancement-based global attention image super-resolution network (EGAN) combining channel- and self-attention mechanisms is proposed for modeling the hierarchical features and intra-layer features in multiple dimensions. Specifically, the channel contrast-aware attention (CCA) module learns the correlations between the intra-layer feature channels and enhances the contrast in the feature maps for richer features in the edge structures. The cyclic shift window multi-head self-attention (CS-MSA) module captures the long-range dependencies between layered features and captures more valuable features in the global information network. Experiments are conducted on five benchmark datasets for $\times 2$, $\times 3$ and $\times 4$ SR. The experimental results show that for $\times 4$ SR, our network improves the average PSNR by 0.12 dB, 0.19 dB and 0.12 dB over RCAN, HAN and NLSN, respectively, and can reconstruct a clear and complete edge structure.

**Keywords:** single-image super-resolution; deep learning; global attention; channel contrast-aware attention; cyclic shift window multi-head self-attention

## 1. Introduction

The single-image super-resolution (SR) task aims to reconstruct degraded low-resolution (LR) images into high-resolution (HR) images with the desired edge structure and texture details. However, the SR is essentially an indeterminate inverse problem where multiple HR images can be generated from the same LR image. In response, numerous SR methods have been proposed, ranging from older interpolation [1] and reconstruction methods [2] to more recent popular learning approaches [3]. Through the ground-breaking introduction of convolutional neural networks (CNNs) into an SR task, a breakthrough was achieved by Dong et al., who proposed an SRCNN [4] network containing three convolutional layers. As a result, CNN-based SR has been widely studied.

The current CNN-based SR networks all use structural cascades or parallelism to increase the network depth. When the number of network layers continues to deepen, overfitting is easily caused owing to the large network size, which makes the model converge slowly or not at all. However, a CNN often needs numerous iterations to generate an optimal solution. Therefore, the key to improving the overall performance of the network is automatically filtering effective and stable features from a large number of data and classifying such features to achieve the mapping of shallow information to deeper layers. On this basis, Kim et al. proposed a VDSR [5] network containing 20 convolutional layers using global residual learning. Meanwhile, an adjustable gradient cropping strategy is applied to accelerate the convergence and avoid a gradient disappearance and explosion. To reduce the number of parameters in the model, Kim et al. proposed a deep recursive convolutional network based on a DRCN [6], which shares convolutional parameters across

layers without introducing additional parameters. To further improve the performance, Lim et al. proposed an enhanced deep convolutional network (EDSR) [7] by improving the residual blocks in SRResNet. Zhang et al. proposed a densely connected RDN [8]-based network to enhance the multi-scale features.

Although significant progress has been made in SR tasks, certain limitations still exist: (1) The previously proposed networks do not fully utilize the large amount of low-frequency information and valuable high-frequency information contained in LR images, resulting in relatively low model performance. (2) The networks proposed in recent years have mainly focused on designing deeper or wider network architectures for studying the reconstruction of the texture details. However, this approach increases the computational burden and reconstructs images with problems such as edge information loss and visual artifacts, leading to excessive image smoothing. Recent studies have found that an attention mechanism can effectively preserve the rich information features and suppress redundant features in LR images, leading to its wide application in SR tasks. However, existing networks apply an attention mechanism in a single dimension to learn the correlation between features. Although this procedure enables the CNN to treat each feature differently in one dimension, features of different depths are treated equally in the global network; thus, redundant features in the network are not completely suppressed. Second, a traditional attention mechanism is unable to focus on all high-frequency texture information present in LR images, resulting in reconstructed images that are still smoothly blurred at the local edge structures. Therefore, reconstructing clear and comprehensive edge texture features has always been an important problem of an SR task.

To address these problems, this paper proposes an SR network (EGAN) based on a global attention mechanism. Among global attention mechanisms, channel contrast-aware attention (CCA) and cyclic shift window multi-head self-attention (CS-MSA) are key to learning the correlation between intra-layer features and hierarchical features. Inspired by 2DDCT, channel attention is based on global average pooling aggregating global features, which are described in frequency domain space with all low- and high-frequency components. Global average pooling can only aggregate the lowest frequency component information in the frequency domain space. Therefore, CCA represents global features as the sum of global average pooling and global standard deviation and enhances the contrast of the feature map to aggregate more high-frequency features. In addition, considering that the correlation between hierarchical features is ignored by using channel attention only, the proposed CS-MSA is applied between hierarchical features to learn long-distance dependencies in global features. Unlike traditional non-local self-attention, CS-MSA simultaneously balances the problems of high computational complexity and the inability to interact with information between windows through window attention and circular shift operations. The main contributions of this research are as follows:

- In this paper, a global attention SR network (EGAN) with joint channel- and self-attentive mechanisms is constructed. The network is capable of exploring correlations between features in terms of intra-layer feature channels and space and between hierarchical feature locations. Experimental results show that the network in this paper outperforms current state-of-the-art networks in most cases with lower complexity.
- As channel attention results in the loss of a large number of high-frequency features present in low resolution, this paper introduces a global adaptive enhancement algorithm to propose channel contrast-aware attention. The contrast of the feature map is enhanced based on global average pooling combined with global standard deviation to effectively aggregate valuable high-frequency features of LR images.
- Since existing methods ignore the correlation between hierarchical features, this paper proposes cyclic shift window attention to consider the correlation between hierarchical features to learn the long-range dependencies in global features. Meanwhile, the introduced cyclic shift and window attention methods effectively solve the problem of the large computational complexity of non-local self-attention.

The remainder of the paper is organized as follows: In Section 2, the main focus is on reviewing current state-of-the-art SR methods with widely used attention mechanisms. In Section 3, the overall structure and theoretical rationale of the proposed method are elaborated. In Section 4, the dataset, experimental setup and evaluation metrics are first detailed, followed by a quantitative and qualitative comparison of the results of the proposed method with state-of-the-art methods. In Section 5, the paper is summarized with an outlook.

## 2. Related Work

SR can be broadly divided into two main categories: traditional methods and deep CNN-based methods. Due to the powerful learning ability of CNNs, the traditional methods have been outperformed by their CNN-based counterparts. In this section, we survey representative works on CNN-based SR and attention-based SR.

### 2.1. Deep CNN-Based Networks

In recent years, CNN-based SR methods have been successfully applied to various tasks and have shown excellent performance. SR can be classified into three main categories according to its implementation: interpolation, reconstruction and learning methods. CNNs have been widely studied due to their strong nonlinear representation capability. Dong et al. [4] first explored the use of three convolutional layers for SR and achieved better results than traditional methods, which laid the foundation for subsequent research. Since then, various fine structures have been proposed, such as residual network structures, iterative inverse projection structures and dense network structures. More and more variant structures have been proposed for application in SR, such as VDSR [5], DRCN [6], EDSR [7] and RDN [8]. Although these methods have achieved good performance, the parameters increase dramatically with the network depth. In addition, these methods treat all features equally, hindering the ability of the network to discriminate between different types of features (low- and high-frequency features).

### 2.2. Attention-Based Networks

An attention mechanism can enable CNNs to focus on valuable regions of feature information and suppress redundant information [9,10]. In recent years, attention mechanisms have been widely used in various computer vision tasks [11,12], significantly improving the performance of networks in various tasks. Hu et al. first proposed the channel attention network (SENet) [9], which significantly improved the feature representation capability of the network by modeling the correlation between feature channels. Wang et al. proposed a non-local block [13] to compute the response of a location to the information of all positions. In SR research, numerous works based on attention mechanisms have been proposed to further improve SR performance. Zhang et al. proposed a deep residual channel-based attention network (RCAN) [14] to learn correlations between features in the channel dimension. Dai et al. proposed a higher-order channel attention (SOCA) [15], which employed second-order feature statistics to learn more discriminative feature expressions. Niu et al. proposed a new holistic attention network (HAN) [16] to model the holistic interdependencies between layers, channels and locations. Mei et al. combined non-local spatial attention with sparse representation and proposed a non-local sparse attention network (NLSN) [17].

Although these methods have achieved significant improvements over CNN-based SR methods, they still have certain limitations: (1) the global averaging pooling operation used in channel attention aggregates only the lowest frequency components under the frequency domain space; (2) traditional non-local networks [13] need to compute the similarity matrix between query feature mappings and key mappings on a pixel-by-pixel basis and thus obtain the long-distance dependencies between all pixels in the feature space; (3) the attention mechanism, as a plug-and-play module at any location in the network, is treated as a locally independent operation that ignores correlations between hierarchical features. To address the above issues, this paper proposes a global attention network that calculates

the importance between intra-layer features and layered features among feature channels, spaces and locations and combines local contrast and window attention to address the shortcomings of channel attention and non-local attention.

### 3. Methods

In this section, the overall structure of EGAN is first detailed, and then CCA and CS-MSA are explored in detail.

### 3.1. Network Structure

The overall network structure of this paper, shown in Figure 1, consists of shallow feature extraction (SFE), deep feature extraction (DFE) and image reconstruction (IR) modules. Given a degraded LR image $I^{LR} \in \mathbb{R}^{3 \times H \times W}$, where H and W are the height and width of the LR image, respectively, the shallow features $F_0 \in \mathbb{R}^{C \times H \times W}$ are first extracted, where C is the number of intermediate feature channels after expansion, denoted as $H_{SFE}$.

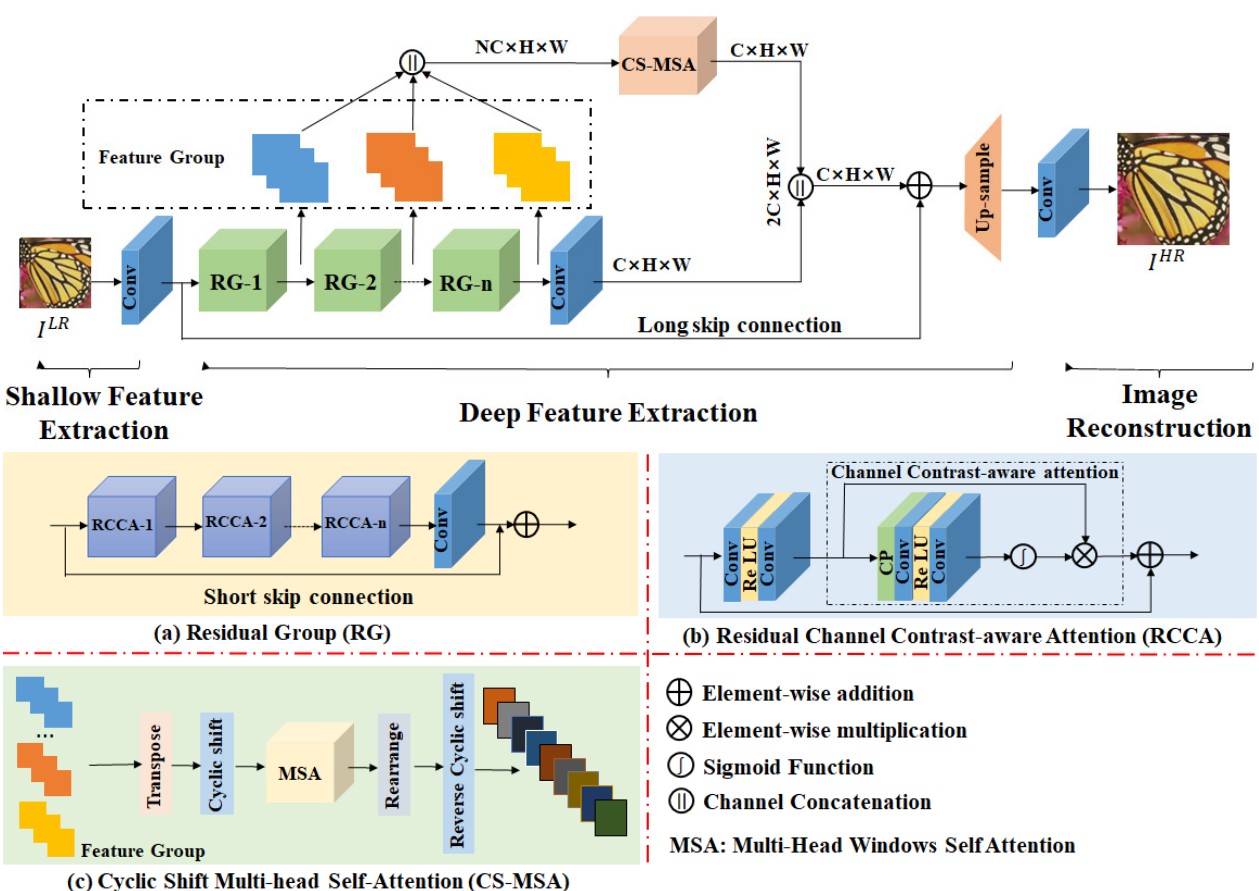

**Figure 1.** Overall structure of the proposed EGAN network.

$$F_0 = H_{SFE}\left(I^{LR}\right) \tag{1}$$

where $F_0$ is transmitted sequentially to the depth feature extraction module, containing $N$ residual groups (RG). One of the RGs is marked as $H_{RG}$.

$$F_i = H_{RG}(F_{i-1}) \ s.t. \ i = 1, 2, \ldots, N \tag{2}$$

where $F_i \in \mathbb{R}^{C \times H \times W}$ is the output feature of the $i$-th RG. Thus, except for $F_N$, which is the final output of the module, all other output features are intermediate. Next, the global features of the network are further weighted, and this process is applied using CS-MSA,

which is denoted as $H_{CS-WSA}$. The module first fuses all intermediate output features, denoted as $H_{LC}$.

$$F_L = H_{CS-MSA}(H_{LC}(concat(F_1, F_2, \ldots, F_N))) \tag{3}$$

where $F_L \in \mathbb{R}^{NC \times H \times W}$ is the weighted fusion feature. The fusion of the features $F_L$ and $F_N$ is achieved through a $1 \times 1$ convolution layer compression channel, and then using the subpixel convolution, the final HR image $I^{HR} \in \mathbb{R}^{3 \times H \times W}$, denoted as $H_{IR}$, is reconstructed.

$$I^{HR} = H_{IR}(concat(F_L, F_N)) \tag{4}$$

### 3.2. Channel Contrast-Aware Attention

Channel attention [9] uses a global average pooling operation to fuse the global information in the channel field and capture valuable regions in the global information through a gating mechanism; this approach helps the network express the global information of the image. However, calculations from the frequency domain show that a global averaging pooling operation is equivalent to the lowest frequency component of the two-dimensional discrete cosine transform (2DDCT) [18]. Specifically, given size $C \times H \times W$ of the input feature $X$, the 2DDCT yields:

$$f_{h,w}^{2d} = \sum_{i=0}^{H-1} \sum_{j=0}^{W-1} X_{i,j}^{2d} \cos(\pi h/H(i+1/2)) \cos(\pi w/H(j+1/2))$$
$$s.t.\, h \in \{0,1,\ldots,H-1\}, w \in \{0,1,\ldots,W-1\} \tag{5}$$

where $\cos(\pi h/H(i+1/2)) \cos(\pi w/H(j+1/2))$ is the basis function of the $2DDCT$, and $f^{2d} \in R^{H \times W}$ is the spectrum of the 2DDCT. Conversely, its two-dimensional inverse discrete cosine transformer (2DIDCT) yields:

$$X_{h,w}^{2d} = \sum_{h=0}^{H-1} \sum_{w=0}^{W-1} f_{i,j}^{2d} \cos(\pi h/H(i+1/2)) \cos(\pi w/H(i+1/2)) \tag{6}$$

$$s.t.\, i \in \{0,1,\ldots,H-1\}, j \in \{0,1,\ldots,W-1\}$$

Expanding the above equation yields:

$$X_{i,j}^{2d} = f_{0,0}^{2d} B_{0,0}^{i,j} + f_{0,1}^{2d} B_{0,1}^{i,j} + \ldots + f_{H-1,W-1}^{2d} B_{H-1,W-1}^{i,j} \tag{7}$$

where $B_{h,w}^{i,j}$ is the basis function of 2DDCT. For Formula (5), when $h$ and $w$ are equal to zero, applying them provides:

$$f_{0,0}^{2d} = \sum_{i=0}^{H-1} \sum_{j=0}^{W-1} X_{i,j}^{2d} \cos(\pi 0/H(i+1/2)) \cos(\pi 0/H(j+1/2)) = gap\left(x^{2d}\right) \tag{8}$$

where $f_{0,0}^{2d}$ can be expressed as the lowest frequency component of 2DDCT and is proportional to the global average pooling result. Finally, bringing the above equation into Equation (7) yields:

$$X_{i,j}^{2d} = gap\left(X^{2d}\right) HW B_{0,0}^{i,j} + f_{0,1}^{2d} B_{0,1}^{i,j} + \ldots + f_{H-1,W-1}^{2d} B_{H-1,W-1}^{i,j} \tag{9}$$

From the above equation, it can be concluded that the global averaging pooling operation retains only the lowest frequency components in the frequency domain space and discards all high-frequency components. Therefore, the CCA module is introduced to retain the valuable high-frequency information in LR images.

The overall structure of a CCA module is shown in Figure 2, where the contrast-aware operation $H_{cp}$ is used to replace the global average pooling. Specifically, CCA combines a local contrast enhancement algorithm [19] with a global average pooling operation to enhance the contrast of a feature map. It is assumed that given a feature map $X$ of size $H \times W$, $X(i, j)$ is a point in the image, and a local window of size $(2n+1) \times (2n+1)$ is delineated with $(i, j)$ as the center, and n is the radius of the window. The low-frequency

information in the local area can be represented by the local mean value, expressed as follows:

$$L_x(i,j) = \frac{1}{(2n+1)^2} \sum_{k=i-n}^{i+n} \sum_{m=j-n}^{j+n} x(k,m) \tag{10}$$

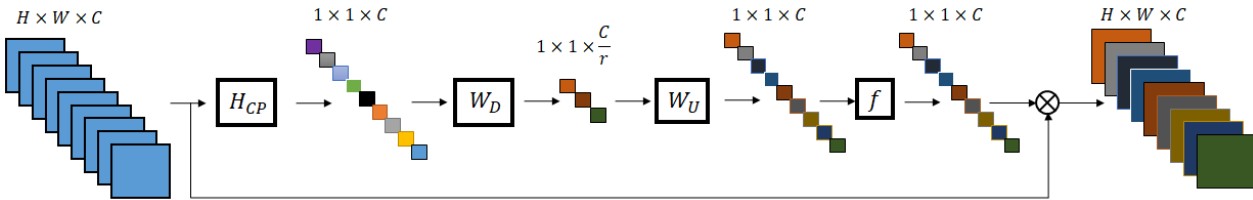

**Figure 2.** Overall structure of the channel contrast-aware attention module.

High-frequency information for local areas is represented by subtracting the low frequencies from the original map as follows:

$$H_x(i,j) = x(i,j) - L_x(i,j) \tag{11}$$

The enhanced feature map consists of the amplified high-frequency information added to the low-frequency information, and the amplification factor is the contrast gain G. $f(i,j)$ is defined as the enhanced feature map, expressed as follows:

$$f(i,j) = L_x(i,j) + G \times H_x(i,j) \tag{12}$$

where $G$ is constant and greater than 1. However, the value of $G$ varies for different images and the effect of enhancement varies. Therefore, we use the inverse of the local standard deviation to represent $G$, making $G$ an adaptive variable. At the edges of the image or where the degree of change is drastic, where the local mean squared difference is large, the value of $G$ is taken to be smaller so that no ringing effect is produced. In smooth areas of the image, where the local mean squared deviation is small, the value of $G$ is taken to be larger, which in turn highlights the contrast of the feature map. The local mean squared deviation is expressed as follows:

$$\sigma_x(i,j) = \sqrt{\frac{1}{(2n+1)^2} \sum_{k=i-n}^{i+n} \sum_{m=j-n}^{j+n} [x(k,m) - L_x(i,j)]^2} \tag{13}$$

In response, the enhanced feature map $f(i,j)$ is represented as follows:

$$f(i,j) = L_x(i,j) + \frac{1}{\sigma_x(i,j)} H_x(i,j) \tag{14}$$

Mapped to a feature map of size $H \times W$, for the C-th channel, $H_{cp}$ can be expressed as:

$$H_{cp}(X_c) = f(X_c) \tag{15}$$

Thus, *CCA* can be expressed as:

$$CCA = f(W_U \gamma(W_D H_{CP})) \tag{16}$$

where $f(\cdot)$ and $\gamma(\cdot)$ denote the sigmoid function and ReLU activation function, respectively; $W_D$ denotes the convolutional weights with channel reduction ratio r; and $W_U$ denotes the convolutional weight with expansion ratio r.

### 3.3. Cyclic Shift Multi-Head Self-Attention Module

In order to improve the deficiencies of the great computational effort of the traditional non-local self-attentive mechanism [13], recent studies [15,20] have divided a number of

fixed-size windows and computed non-local attention within each window separately. Although the number of participants is greatly reduced, information interaction between windows is not possible. Benefiting from [21], this paper optimizes this procedure based on the sliding window mechanism to achieve information interaction between windows with a circular shift mechanism. A cyclic shift window self-attentive is proposed to adaptively learn the long-distance dependence between hierarchical features. In addition, the circular shift mechanism removes the masking strategy and relative position coding, making the network structure more streamlined and efficient.

The overall process of a cyclic shift window is shown in Figure 3. Given the size $H \times W \times C$ of the input feature A, and assuming this feature is divided into local non-overlapping $M \times M$ windows, A is circularly shifted toward the bottom and right positions by M/2, respectively. The local multi-head self-attention (MSA) for every window is then calculated after the shifting of positions of the feature maps. Finally, through a reverse cyclic shift, we obtain feature map B by shifting its position up and to the left by M/2. Each window in feature map B introduces the information in the adjacent non-overlapping windows and achieves a global information interaction across windows.

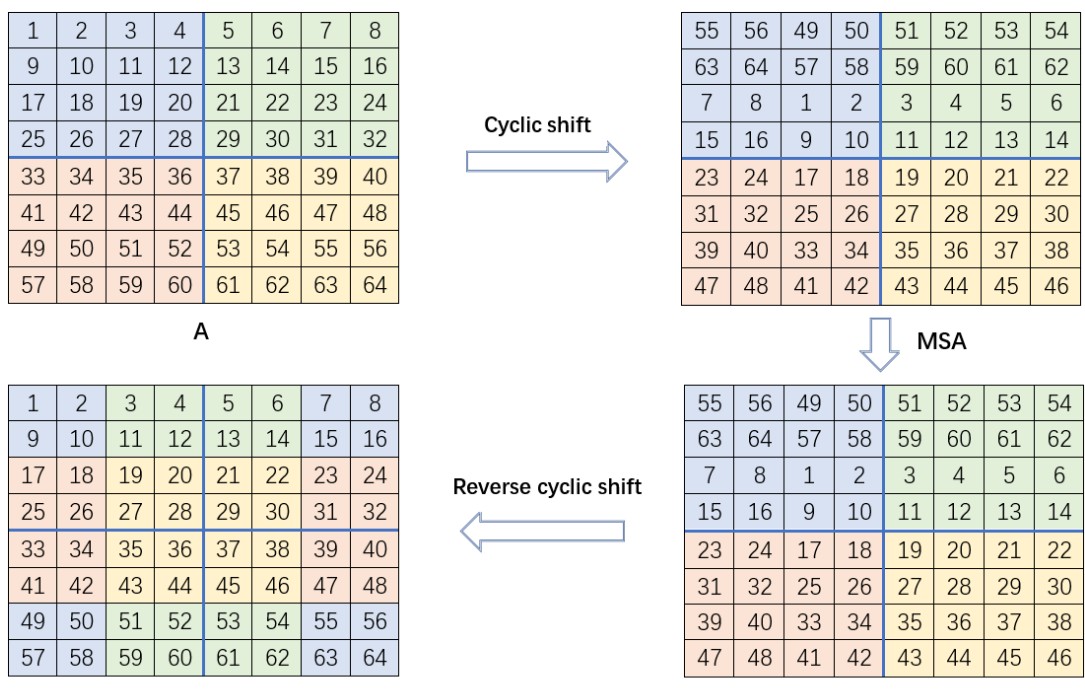

**Figure 3.** Visualization of the process of cyclic shift window multi-head self-attention. A is the input feature map and B is the attention feature map.

The overall structure of the MSA is shown in Figure 4. The input to the module is an intermediate feature extracted from N RGs and is fused as the input feature $F_N$ in the channel dimension. First, we shift the input feature $F_N$ cyclically by M/2 bits along the diagonal direction and then reshape the shifted feature as $B \times M^2 \times NC$; i.e., the shifted features are divided into $B$ mutually non-overlapping $M \times M$ local windows. Here, $B$ represents $HW/M^2$ and denotes the total number of windows. The self-attention is then computed in each window separately. For the local window feature, $X \in \mathbb{R}^{M^2 \times NC}$, we use two linear transformations, Q and V, to map the input feature, X, into the query and the value matrix space.

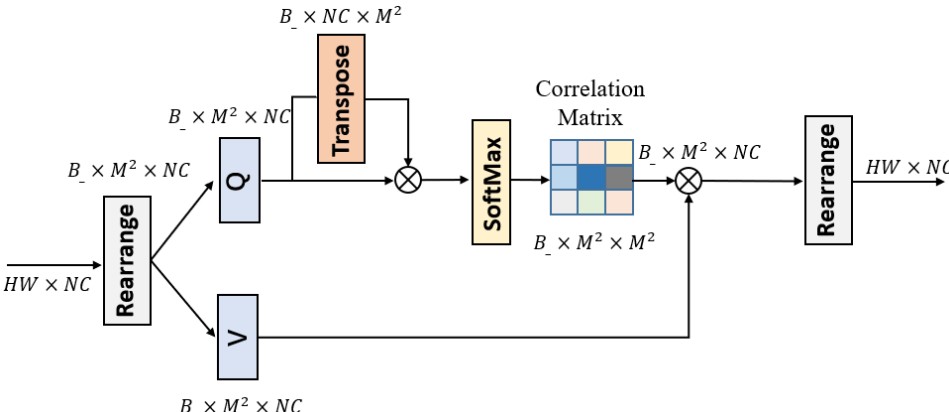

**Figure 4.** Overall structure of the MSA module.

Compared to a traditional self-attention module, the MSA utilizes a shared key instead of the query key Q and the queried key K to compute the self-attention in a symmetric Gaussian space. This approach is also shown to further reduce the computational burden of self-attention without affecting performance [22]. Unlike non-local networks where query, key and value perform a single attention function, in MSA, the query and key perform the attention function h times in parallel, and the results are fused for multi-head attention. Thus, the MSA computation process is expressed as follows:

$$Attention(Q, V) = SoftMax\left(QQ^T / \sqrt{d}\right) V \tag{17}$$

where $d$ represents the dimension of query/key.

## 4. Experiment

In this section, we first detail the dataset used in the experiments and its parameter settings and then present ablation experiments conducted on the two proposed core modules to verify their effectiveness. Secondly, the overall strength of the proposed network is verified using five benchmark test sets. Finally, the network performance is analyzed by showing the training curves and execution times.

### 4.1. Datasets and Performance Metrics

For training, we use 800 training images from the DIV2K [23] dataset as the training set. DIV2K is a publicly available and commonly used high-quality (2K resolution) image dataset mainly used for image recovery tasks. For testing, five publicly available benchmark test sets were selected: Set5 [24], Set14 [25], BSDS100 [26], Urban100 [27] and Manga109 [28]. Among them, Set5, Set14 and BSDS100 consist of natural images with rich texture details, and Urban100 and Manga109 consist of images of urban landscapes and anime characters with rich edge structures.

In addition, we leveraged the peak signal-to-noise ratio (PSNR) [29] and the structural self-similarity (SSIM) [30] as quantitative evaluation metrics for the performance of the final reconstructed SR images. Since the human eye is most sensitive to luminance, we only calculate $PSNR$ and $SSIM$ values on the Y channel of the YCbCr channel. Given a ground-truth image $I^{HR}$ and an SR image $I^{SR}$, the PSNR can be defined as:

$$PSNR\left(I^{HR}, I^{SR}\right) = 10 \, log_{10}\left(\frac{L^2}{MSE}\right) \tag{18}$$

where

$$MSE = \frac{1}{HW} \sum_{i=1}^{H} \sum_{j=1}^{W} \left(I^{HR}(i,j) - I^{SR}(i,j)\right)^2 \tag{19}$$

where $L$ denotes the maximum pixel value of the image, which is generally defined to be 255. $H$ and $W$ are the height and width, respectively. The higher the $PSNR$ value, the better the image fidelity and the higher the quality of the reconstruction. The $SSIM$ can be defined as:

$$SSIM\left(I^{HR}, I^{SR}\right) = \left[L\left(I^{HR}, I^{SR}\right)\right]^{\alpha} \left[C\left(I^{HR}, I^{SR}\right)\right]^{\beta} \left[S\left(I^{HR}, I^{SR}\right)\right]^{\gamma} \tag{20}$$

where

$$L\left(I^{HR}, I^{SR}\right) = \frac{2\mu_{I^{HR}}\mu_{I^{SR}} + b_1}{\mu_{I^{HR}}^2 + \mu_{I^{SR}}^2 + b_1}$$

$$C\left(I^{HR}, I^{SR}\right) = \frac{2\sigma_{I^{HR}}\sigma_{I^{SR}} + b_2}{\sigma_{I^{HR}}^2 + \sigma_{I^{SR}}^2 + b_2} \tag{21}$$

$$S\left(I^{HR}, I^{SR}\right) = \frac{2\sigma_{I^{HR}I^{SR}} + b_3}{\sigma_{I^{HR}}\sigma_{I^{SR}} + b_3}$$

where $\alpha$, $\beta$ and $\lambda$ are the adjustment parameters that control the relative importance of the three comparison functions, respectively. $\mu$ denotes the mean calculation, and $\sigma$ denotes the variance calculation. In addition, $b_1$, $b_2$ and $b_3$ are all constants used to avoid a denominator of zero. The calculated SSIM takes values within the range of [0, 1]. The higher the value, the lower the image distortion and the better the reconstruction.

### 4.2. Settings

In this paper, the RG number, RCCA number, window size, channel number and attention head number are set to 6, 20, 8, 64 and 4, respectively. For ×2, ×3 and ×4 training, we obtained the input LR images from the corresponding HR images by bicubic downsampling in the training stage. Then, we set 16 LR patches as each training mini-batch and performed extraction with a size of 48 × 48 from the LR images. Moreover, we randomly rotated the image in the training dataset by 90°, 180° and 270° and flipped it horizontally for data augmentation. We utilized Adam optimizer [31] to optimize our model with settings of β1 = 0.9 and β2 = 0.999. We fixed the initial learning rate to $10^{-4}$ and decreased the learning rate by half every 400 epochs. We implemented the proposed model on the PyTorch [32] framework with an NVIDIA TITAN XP GPU.

### 4.3. Comparison with State-of-the-Art Methods

To verify the effectiveness of the proposed networks, six advanced SR networks are selected for comparison: BICUBIC [33], EDSR [7], RDN [8], RCAN [14], SAN [15], HAN [16] and NLSN [17].

For the fairness of the experiments, we retrained all algorithms according to the experimental setup described in the previous section. Table 1 provides quantitative results for scale factors of ×2, ×3 and ×4 on the five benchmark datasets. It can be seen that our proposed network outperforms the seven networks on different datasets and scale factors. Specifically, compared to two CNN architecture-based networks (EDSR and RDN), the network in this paper achieved higher PSNR and SSIM values on different datasets and scale factors. Although MSRN has slightly fewer parameters than the proposed network, its reconstruction results are far worse. Compared to the four attention-based CNN networks (RCAN, SAN, HAN and NLSN), the present network achieves higher performance with fewer parameters (Params and FLOPs). In particular, this network achieves a further improvement over SAN and HAN, which also use the same RCAN backbone network. Specifically, when reconstructing the results for the Set5 dataset at scale ×4, this network achieves a 0.22 dB improvement in PSNR compared to SAN and a 0.22 dB improvement compared to HAN. When reconstructing the results for the Urban100 dataset at scale x4, this network achieves a 0.22 dB improvement in PSNR compared to RCAN and a 0.18 dB improvement compared to HAN. In addition, the network achieves comparable performance with a lower number of parameters compared to the NLSN. The improvement

in the PSNR metric is 0.04 dB when reconstructing the results on the Urban100 dataset at x4 scale and 0.31 dB when reconstructing the results on the Manga109 dataset at $\times 4$ scale.

**Table 1.** Quantitative comparison with advanced methods for classical image SR on a benchmark test set (mean PSNR/SSIM). The CNN-based methods and attention-based methods are separated by a dashed line for each scaling factor. "-" means that the result is not available. "NaN" means that the current device cannot be tested. The best PSNR/SSIM indexes are marked in red and blue colors, respectively. Note that all the efficiency proxies (Params and FLOPs) were measured under the setting of upscaling SR images to $1024 \times 1024$ resolution on all scales.

| Methods | Scale | Params (K) | FLOPs (G) | Set5 PSNR/SSIM | Set14 PSNR/SSIM | BSDS100 PSNR/SSIM | Urban100 PSNR/SSIM | Manga109 PSNR/SSIM |
|---|---|---|---|---|---|---|---|---|
| BIUCBIC | $\times 2$ | - | - | 33.66/0.9299 | 30.24/0.8688 | 29.56/0.8431 | 26.88/0.8403 | 30.80/0.9339 |
| EDSR | $\times 2$ | 40,730 | 3184 | 36.11/0.9302 | 32.92/0.9095 | 30.32/0.9013 | 30.93/0.8951 | 37.10/0.9773 |
| RDN | $\times 2$ | 22,123 | 1298 | 36.08/0.9305 | 32.74/0.9070 | 30.23/0.8913 | 30.22/0.8826 | 37.82/0.9668 |
| RCAN | $\times 2$ | 15,445 | 1004 | 37.77/0.9598 | 33.43/0.9157 | 32.01/0.8977 | 31.46/0.9219 | 38.18/0.9759 |
| SAN | $\times 2$ | 15,861 | 1012 | 37.80/0.9599 | 33.43/0.9157 | 32.02/0.8981 | NaN | NaN |
| HAN | $\times 2$ | 15,924 | 1035 | 37.81/0.9599 | 33.45/0.9158 | 32.03/0.8980 | 31.51/0.9225 | 38.17/0.9760 |
| NLSN | $\times 2$ | 41,796 | 2740 | 37.83/0.9599 | 33.44/0.9158 | 32.01/0.8976 | 31.44/0.9217 | 38.20/0.9759 |
| Our method | $\times 2$ | 9672 | 614 | 37.84/0.9602 | 33.46/0.9159 | 32.05/0.8981 | 31.58/0.9227 | 38.33/0.9766 |
| BIUCBIC | $\times 3$ | - | - | 30.41/0.8655 | 27.64/0.7722 | 27.21/0.7344 | 24.46/0.7411 | 26.96/0.8555 |
| EDSR | $\times 3$ | 43,680 | 3276 | 33.80/0.9213 | 29.92/0.8339 | 28.80/0.7981 | 27.28/0.8320 | 32.28/0.9338 |
| RDN | $\times 3$ | 22,308 | 1475 | 33.73/0.9211 | 29.90/0.8332 | 28.79/0.7972 | 27.10/0.8276 | 32.35/0.9335 |
| RCAN | $\times 3$ | 15,629 | 1017 | 33.82/0.9223 | 29.99/0.8403 | 28.84/0.7982 | 27.38/0.8314 | 32.36/0.9348 |
| SAN | $\times 3$ | 15,897 | 1024 | 33.90/0.9232 | 30.01/0.8310 | 28.89/0.7988 | NaN | NaN |
| HAN | $\times 3$ | 16,109 | 1048 | 34.11/0.9242 | 30.14/0.8369 | 28.91/0.8001 | 27.56/0.8387 | 32.73/0.9379 |
| NLSN | $\times 3$ | 44,747 | 2935 | 34.15/0.9249 | 30.12/0.8367 | 28.92/0.8003 | 27.62/0.8404 | 32.84/0.9392 |
| Our method | $\times 3$ | 9856 | 626 | 34.12/0.9252 | 30.12/0.8369 | 28.94/0.8013 | 27.63/0.8407 | 33.03/0.9399 |
| BIUCBIC | $\times 4$ | - | - | 28.43/0.8022 | 26.10/0.6936 | 25.97/0.6517 | 23.14/0.6599 | 24.91/0.7826 |
| EDSR | $\times 4$ | 43,090 | 3294 | 31.72/0.8880 | 28.28/0.7741 | 27.36/0.7288 | 25.39/0.7628 | 29.44/0.8933 |
| RDN | $\times 4$ | 22,271 | 1490 | 31.63/0.8864 | 28.20/0.7719 | 27.31/0.7272 | 25.30/0.7600 | 29.46/0.8924 |
| RCAN | $\times 4$ | 15,592 | 1044 | 31.80/0.8891 | 28.34/0.7749 | 27.39/0.7300 | 25.46/0.7666 | 29.75/0.8970 |
| SAN | $\times 4$ | 15,861 | 1059 | 31.79/0.8887 | 28.31/0.7748 | 27.38/0.7298 | NaN | NaN |
| HAN | $\times 4$ | 160,71 | 1075 | 31.79/0.8898 | 28.32/0.7753 | 27.40/0.7307 | 25.50/0.7682 | 29.73/0.8976 |
| NLSN | $\times 4$ | 44,157 | 3364 | 31.91/0.8902 | 28.36/0.7753 | 27.41/0.7305 | 25.64/0.7698 | 29.81/0.8985 |
| Our method | $\times 4$ | 9820 | 654 | 32.01/0.8915 | 28.40/0.7771 | 27.46/0.7325 | 25.68/0.7734 | 30.12/0.9015 |

In addition, we compared the visual quality from the Set14, BSDS100, Urban100 and Manga109 datasets at a scale factor of $\times 4$. As shown in Figure 5, for "zebra" from Set14 and "8023" from BSDS100, we observed that most algorithms recovered visible edges, but with different degrees of blurring. This network continuously reconstructs the stripes on zebra and bird, while EDSR, SAN and HAN produce incorrect textures. In addition, the stripe structure recovered by this network is brightly colored and closer to the original image. For "img_073" from Urban100 and "YumeriroCooking" from Manga109, we observed that the other methods had varying degrees of artifacts, producing relatively blurred lines, incorrect textures and dull colors. In contrast, the network proposed in this paper was able to reconstruct the stripes on the building and the fabric in full, and the color effects were much closer to the original image.

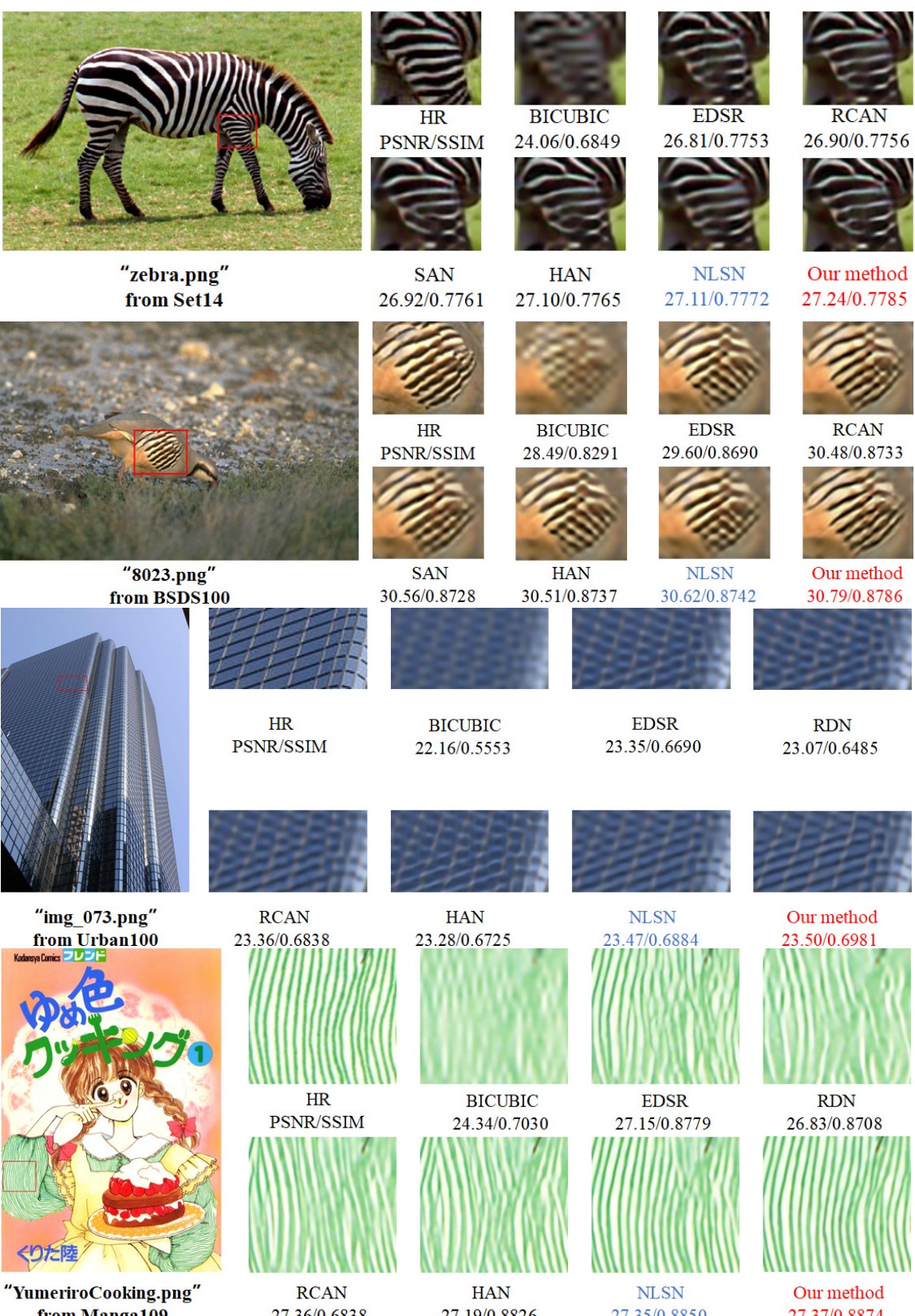

**Figure 5.** Visual comparison results of our method with EDSR, RDN, RCAN, SAN, HAN and NLSN for ×4 SR images on Set14, BSDS100, Urban100 and Manga109 datasets. Best PSNR/SSIM indexes are marked in red and blue colors, respectively.

### 4.4. Ablation Studies

In this section, we present the results of multiple sets of ablation experiments conducted to validate the effectiveness of the CCA and CS-MSA modules. To save training time, all comparison experiments were trained 100 times at ×2 scale factors to evaluate performance. The baseline model consists of 6 RGs and 20 basic residual blocks, with no attention blocks in the basic residual blocks. Channel attention (CA), channel contrast-aware attention (CCA), window self-attention (WSA) and cyclic shift window multi-head self-attention (CS-MSA) blocks were added to this base for ablation experiments.

#### 4.4.1. Combination with CCA

The effectiveness of the CCA module was investigated, and the results are shown in Table 2. It can be seen that the integrated CA module improves the average PSNR by 0.48 dB over the five benchmark datasets compared to the baseline model, which validates that the channel attention mechanism can effectively improve the performance of the SR network. The network performance is significantly improved by the integration of the CCA module, with an average PSNR improvement of 0.11 dB compared to the CA module. This is due to the fact that the CCA module combines local standard deviation to improve the contrast of the feature maps and effectively aggregates more edge texture information.

**Table 2.** Effectiveness of CCA on benchmark datasets for ×2 SR. Best PSNR/SSIM indexes are marked in red and blue colors, respectively.

| Methods | Set5 PSNR/SSIM | Set14 PSNR/SSIM | BSDS100 PSNR/SSIM | Urban100 PSNR/SSIM | Manga109 PSNR/SSIM |
|---|---|---|---|---|---|
| Baseline | 36.29/0.9508 | 32.22/0.9034 | 31.12/0.8845 | 28.94/0.8857 | 35.13/0.9614 |
| Baseline + CA | 36.70/0.9527 | 32.44/0.9047 | 31.30/0.8858 | 29.55/0.8954 | 36.09/0.9660 |
| Baseline + CCA | 36.80/0.9541 | 32.56/0.9074 | 31.40/0.8889 | 29.68/0.8981 | 36.21/0.9676 |

To demonstrate the effect of contrast more visually, this section visualizes the output feature maps for both models located at a shallow level with the integrated CA module and the CCA module. As seen in Figure 6, the feature maps are clearer and the edge structure is more prominent with the integration of the CCA module.

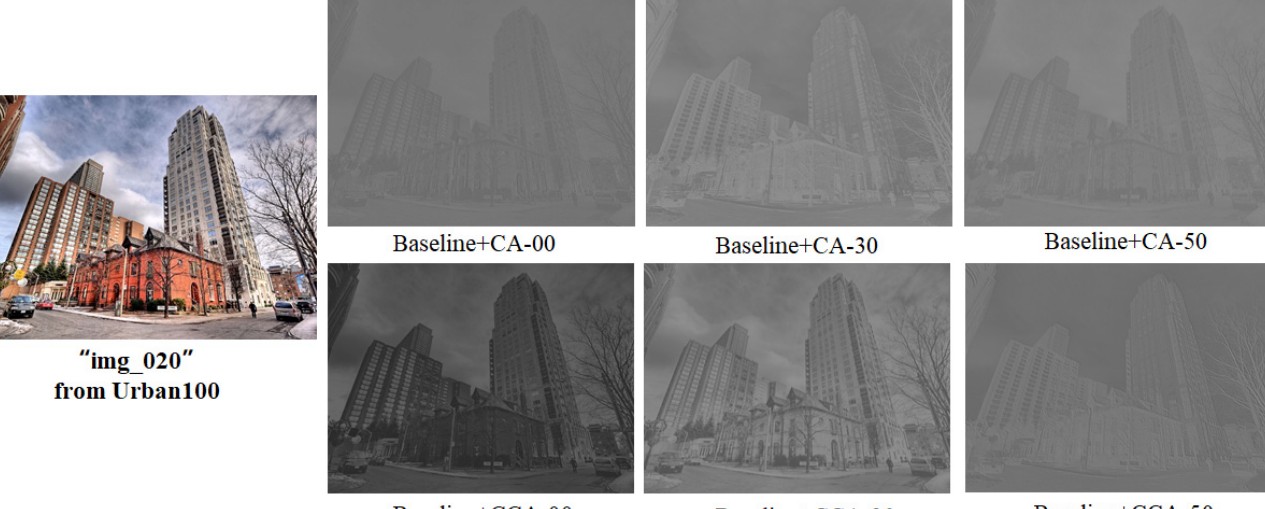

"img_020" from Urban100

Baseline+CA-00    Baseline+CA-30    Baseline+CA-50

Baseline+CCA-00    Baseline+CCA-30    Baseline+CCA-50

**Figure 6.** Residual block visualization output feature maps. Here, 00 denotes the first residual block in the first residual group in the network.

### 4.4.2. Combination with CS-MSA

To verify the effectiveness of the circular shift mechanism, this section compares it with the window mechanism and the sliding shift window mechanism. These three mechanisms all use MSA to perform non-local attention and are named window multiheaded attention (W-MSA), sliding shift window multiheaded attention (SW-MSA) and the proposed circular shift window multiheaded attention (CS-MSA), respectively. Table 3 reports the PSNR/SSIM metrics for the three benchmark datasets as well as the number of parameters (Params), floating point computations (FLOPs) and inference times (Runtimes). Specifically, we integrate each of these three modules from Baseline + CCA. As expected, the average PSNR and SSIM of the three datasets improved by 0.1 dB and 0.0012, respectively, after the W-MSA module was integrated, indicating that global information interactivity between learning hierarchy features can significantly improve the performance of the network. The average PSNR and SSIM for the three datasets improved by 0.46 dB and 0.0016, respectively, when SW-MSA was integrated, and Params, FLOPs and Runtimes all improved. We then replaced SW-MSA with CS-MSA, and the average PSNR and SSIM metrics decreased by 0.05 dB and 0.001 for the BSDS100 and Urban100 datasets, respectively, and improved by 0.04 dB and 0.0003 for the Manga109 dataset, respectively. In addition, CS-MSA achieves the same network size, and the average PSNR and SSIM metrics improve by 0.2 dB and 0.0023, respectively, compared to W-MSA. This indicates that CS-MSA can efficiently interact with the information between windows and that the cyclic shift operation does not generate additional parametric quantities.

**Table 3.** Performance comparison of different components on three benchmarks. Best PSNR/SSIM indexes are marked in red and blue colors, respectively. "$\sqrt{}$" indicates the currently used component.

| Methods | Different Components | | | Params | FLOPs | BSDS100 | Urban100 | Manga109 |
|---|---|---|---|---|---|---|---|---|
| | W-MSA | SW-MSA | CS-MSA | (K) | (G) | PSNR/SSIM | PSNR/SSIM | PSNR/SSIM |
| Baseline | $\sqrt{}$ | | | 9902 | 610 | 31.41/0.8893 | 29.77/0.9003 | 36.32/0.9685 |
| + | | $\sqrt{}$ | | 10,032 | 615 | 31.55/0.8901 | 29.98/0.9041 | 36.45/0.9690 |
| CCA | | | $\sqrt{}$ | 9902 | 610 | 31.50/0.8902 | 29.91/0.9021 | 36.49/0.9693 |

### 4.5. Model Analysis

#### 4.5.1. Training Process Curve

This section shows the curve trends of the training loss and validation metrics of the network. Four CNN networks based on the attention mechanism are selected for comparison with the network in this paper, where the validation set is the Set5 dataset. At a magnification factor of 4, the training loss curve is shown in Figure 7a, and the training accuracy curve is shown in Figure 7b. It can be intuitively seen from the curve trends that the network in this paper converges at a more stable rate than the other networks during the training process and achieves a higher validation metric, which stays around 29.01–29.03 dB in the later stages.

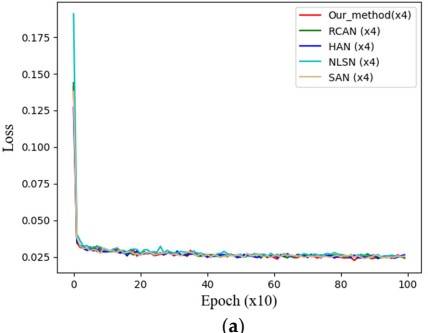 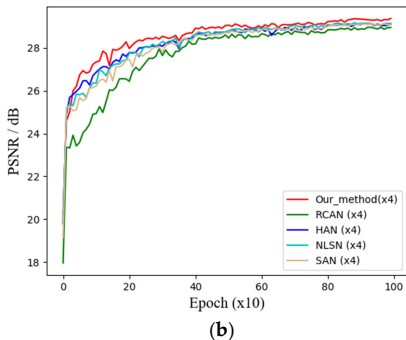

(a)      (b)

**Figure 7.** Training process curve at ×4 SR. (**a**) Training loss curve; (**b**) training accuracy curve.

4.5.2. Execution Time

To fully measure the performance of the network proposed in this paper, model execution time experiments were conducted, and the execution time of the network proposed in this paper was compared with that of other methods. Figure 8 shows the results of the experiments, where the execution times of all models were tested on the 2.10 GHz Intel(R) Xeon(R) Silver 4110 CPU with 48 G RAM. From the figure we observe that the network in this paper achieves the highest performance at the expense of some execution time. In summary, our method offers a better balance between model complexity and reconstruction performance.

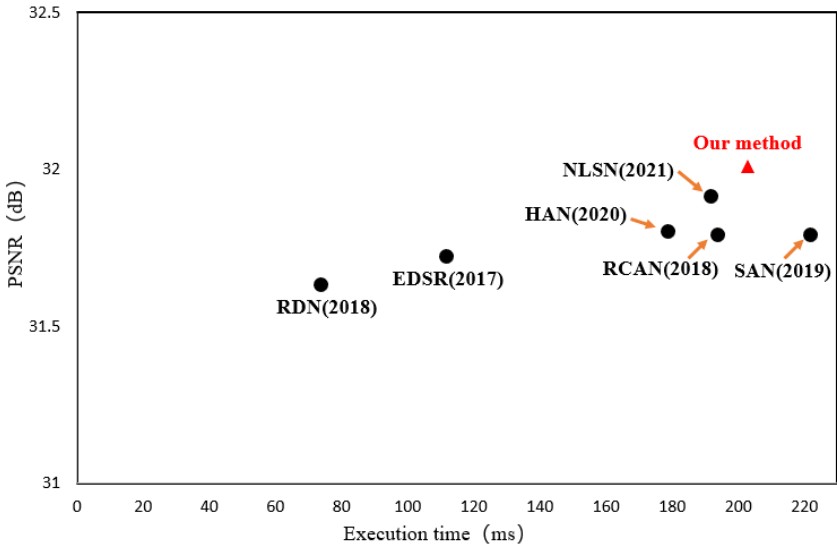

**Figure 8.** Trade-off between performance and execution time on Set5 for ×4 SR. Note that the input resolution is 256 × 256.

**5. Conclusions**

In this paper, we proposed a global attention network for nature image edge enhancement. The network combines channel- and self-attention mechanisms to adaptively learn the global dependencies between layered and intra-layer features in multiple dimensions. Specifically, the CCA module learns the correlation between feature channels within layers and combines the global standard deviation to enhance the contrast of the feature map and enrichen feature edge structure information. The CS-MSA module captures the long-range dependencies between layered features, capturing more valuable features in the global information. Experiments conducted on benchmark datasets at ×2, ×3 and ×4 show that the network in this paper outperforms current state-of-the-art SR networks in terms of performance metrics and visual quality. In particular, the metrics are significantly improved, and the reconstructed image edge structure is clear in the Urban100 and Manga109 datasets where the edge structure is rich. In future work, we will extend the network in this paper to real-world super-resolution tasks, making it capable of resolving real-world degraded images in general.

**Author Contributions:** Conceptualization, Z.S. and W.S.; methodology, Z.Z. and K.N.; software, Z.Z.; validation, Z.Z.; formal analysis, Z.Z. and K.N.; investigation, Z.S., W.S. and K.N.; data curation, Z.Z.; writing—original draft, Z.Z.; writing—review and editing, Z.S., W.S. and K.N. All authors have read and agreed to the published version of the manuscript.

**Funding:** This research received no external funding.

**Institutional Review Board Statement:** Not applicable.

**Informed Consent Statement:** Not applicable.

**Data Availability Statement:** The data presented in this study are publicly available data (sources stated in the citations). Please contact the corresponding author regarding data availability.

**Acknowledgments:** Thanks to all editors and reviewers.

**Conflicts of Interest:** The authors declare no conflict of interest.

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
