# Peer review of "Global Attention Super-Resolution Algorithm for Nature Image Edge Enhancement"

_sustainability, doi:10.3390/su142113865_

Round 1

Reviewer 1 Report

An edge-enhancement-based global attention image super-resolution network (EGAN) combining channel- and self-attention mechanisms is proposed for modelling the hierarchical features and intra-layer features  in multiple dimensions.  The results of extensive experiments show that the proposed network outperforms current state-of-the-art SR networks. But I still have some problems.

1, EGAN  learns  the global features for a natural image edge enhancement, which combines channel- and self-attention mechanisms.  why not use the local features?

2, What's the difference between "EGAN "  with "CBAM"?

3,Why does  "model the intra-layer and layered features"  and how does it work?

4, "based on nature image edge enhancement'  ... may be "for nature image edge enhancement"

5, "a break through was achieved by Dong et al., who proposed an SRCNN network containing three convolutional layers." should be cited.

6, "or the Cth channel, ??? can be expressed as" should be "C-th".

7, what does mean of gap in the Formula 10.

Author Response

Thank you very much for your review of ‘’Global attention super-resolution algorithm based on  nature image edge enhance-ment’’. I have provided detailed answers to the questions raised by the reviewers and have revised the paper in detail in accordance with the reviewers' comments, with the changes marked in red. Response comments are presented in a word document.

Reviewer 2 Report

NOTES:

- Why did you adopt this network structure? Based on what? 

- Is your structure completely new? What is new and what is old?

- Sow also your network training loss and accuracy

- Do not explain well-known methods. Explain what is "different", the innovations

- Why did you choose those methods for comparison? You are comparing your system with the 2007 and 2014 methods

ABSTRACT:

- "pracstical" - Line 9

- "...in this paper achieves a peak signal-to-noise ratio (PSNR) of 26.80 dB at a scale factor of x4..." - Compare your approach with the state-of-the-art methods. Quantify it using e.g. percentage

INTRODUCTION:

- Missing a paragraph describing the article organization "In Section XXX, it will be ....."

- Specify what is "old" and what is "new" in your contributions. What are the innovations

RELATED WORK:

- Start the section by explaining the subsections

METHODS:

- Why did you adopt this network structure? Based on an existing one? Explain your decisions

- Again, what is "old" and what is "new" in this approach - Explain clearly

- Increase the resolution of your figures

- Too many details explaining well-known methods. Just present your innovation and cite the well-known methods e.g. 2DCCT

- Figure 3 - WAM is only defined after you refer to the figure in the text

- Cyclic Shift Window Self-Attention Module is entirely new? Is it not based on anything? 

EXPERIMENTS:

- After the equations, you do not apply indentation since it is the same paragraph

- "Evaluation indicators" - Probably "Performance metrics"

- Missing citations

- Change EGAN to "Our method"

- The overall performance improvement of your method is very low. What is the advantage of your method?

- Processing time? Real-time analysis? Is it faster when compared with other methods?

- Comparison with state-of-the-art methods? Are those the best currently known methods?

- BICUBIC - 2007 / SRCNN - 2014 / LapSRN - 2017 / MSRN - 2018 / EDSR - 2017 / DBPN - 2018 / RCAN - 2018. - Why perform the comparison with the 2007 and 2014 methods? The most recent and better methods are these?

CONCLUSIONS:

- Future work?

- Increase of performance when compared with the other methods? Using e.g. percentage

- The overall performance improvement of your method is very low. What is the advantage of your method?

Author Response

Thank you very much for your review of ‘’Global attention super-resolution algorithm based on  nature image edge enhance-ment’’. I have provided detailed answers to the questions raised by the reviewers and have revised the paper in detail in accordance with the reviewers' comments, with the changes marked in blue. Response comments are presented in a word document

Reviewer 3 Report

In this paper, Image edge and metrics are significantly imporved. Thanks.

Author Response

Thank you very much for your review of ‘’Global attention super-resolution algorithm based on  nature image edge enhance-ment’’. Response comments are presented in a word document

Round 2

Reviewer 2 Report

The authors have addressed all my comments. The article is a small contribution to the field but can be considered one.

Abstract:

- Line 9 - "pracstical" - practical?

- Line 20 - SR? What is SR? Please define the acronym

Introduction:

- Line 86 - "...this research are as follows." should be "...follows:"

Related Work:

- Please start the section by introducing the Subsections
